# Pharmacological Modulation of Host Immunity with Hen Egg White Lysozyme (HEWL)—A Review

**DOI:** 10.3390/molecules28135027

**Published:** 2023-06-27

**Authors:** Alberta Bergamo, Gianni Sava

**Affiliations:** 1Department of Life Sciences, University of Trieste, 34127 Trieste, Italy; abergamo@units.it; 2Italian Society of Pharmacology, 20129 Milano, Italy

**Keywords:** lysozyme, immunity, pharmacology, experimental, human, therapy

## Abstract

In the 100 years since its discovery, lysozyme has become an important molecule, both as model for studies in different fields and as a candidate for the therapy of various pathological conditions. Of the dozens of known lysozymes, in this review we focus on one in particular, lysozyme extracted from hen egg white (HEWL), and its interaction with the immune system when it is administered orally. Experimental data show that there is an axis that directs immune system activation from GALT (gut-associated lymphoid tissue) and the intestinal lymphocyte clusters. Although a contribution of peptidoglycans from digestion of the bacterial cell wall in the intestinal lumen cannot be excluded, immune stimulation is not dependent on the enzymatic activity of HEWL. The immune responses suggest that HEWL is able to recover from immunodepression caused by tumor growth or immunosuppressants, and that it also improves the success of chemotherapy. The positive results obtained in a small Phase 2 study in patients, the ease of oral administration of this protein, and the absence of adverse effects suggest that HEWL may play an important role in all diseases where the immune system is weakened or where its enhancement plays a critical role in the resolution of the pathology.

## 1. Introduction

Scientific knowledge of lysozyme and its physiological role in humans, dating back to early work of its discoverer, Prof. Alexander Fleming in 1922, confirms its protective role against potential bacterial infections [1,2,3,4]. Originally discovered in human nasal mucus, it has subsequently been identified in many body areas that are in direct contact with the external environment, including, notably, the upper respiratory tract and the entire gastrointestinal tract, but also the eye and skin. On this basis, lysozyme, which has also been identified in many other living organisms of the animal and plant kingdoms [5], among which the white of the hen’s egg stands out (Figure 1) [6,7], became an antibacterial agent as a drug to fight infections in humans [8,9,10]. Its administration has been shown to cause allergic phenomena in individuals allergic to eggs and their products, especially when the drug is used in injectable form, as shown in [11] and the references cited therein.

This is where the role of lysozyme in the host immune system comes into play. The interactions of lysozyme with the immune system have been the subject of numerous studies since the 1970s (reference [12] is one example among many others). Many of these studies have attempted to answer the question of why lysozyme elicited an allergic response in the subjects to whom it was administered. In this regard, there is an extensive literature analyzing the lysozyme molecule on the basis of its epitopes and attributing to them the responsibility for the immune reactions against the molecule [13,14,15]. In summary, lysozyme is capable of eliciting an immune response in allergic individuals consisting of reactions against specific epitopes of the molecule itself, which in short correspond to the appearance of specific antibodies.

Many other studies on lysozyme also concerned its ability to induce the immune system to cooperate against infectious diseases. In summary, their main objective was to determine whether the administration of lysozyme has the potential to elicit responses from different compartments of the immune system and whether these responses may be of some therapeutic value, as described in the review by Sun and coworkers [16], who looked at various aspects of the potential therapeutic use of lysozyme, but in their analysis often mixed the biological effects of endogenous lysozymes with those obtained by pharmacological administration of different types of lysozymes from different sources (recombinant human, microorganisms, insects, plants, as well as hen egg white lysozyme (HEWL)), which share some biological characteristics, although these cannot be completely superimposed. In analyzing the mechanism of action of lysozyme in terms of its immunomodulatory potential, the review by Sun and coworkers highlights how the immunomodulatory effect can attenuate different types of inflammatory phenomena, mainly thanks to the interaction with mechanisms supported by TNF-α and interleukin-6 [17,18,19,20].

In the last two decades, a number of excellent papers and reviews on lysozyme have been published, dealing with its antimicrobial activity, its application in the food and feed industry, its allergenic effects, but also its potential for the development of materials based on natural polymers or methods to improve its detection to name but a few [21,22,23,24,25,26].

From the analysis of this extensive literature retrieved from the main scientific databases under the keywords *lysozyme*, *hen egg white lysozyme*, *immunity* and *immunomodulation*, we have filtered out the articles of interest.

The present article aims to review the literature and understand whether and how lysozyme from hen egg white (HEWL) can enhance immunity in animals and whether this enhancement has therapeutic significance in pathologies that are sensitive to host immune responses. Aside from studies in various animal models and with different therapeutic targets, the research community has little information on the effect of HEWL on the human immune system, with the exception of a limited, uncontrolled phase II study in oncology patients, which is briefly referenced at the end of this review.

## 2. Evidence of HEWL Effects on Immunity

Evidence that the pharmacological effects of HEWL are due to its role as a modulator of the immune system comes from the extensive literature reporting its use as an anticancer agent. Without reproducing in detail what is reported in the review entitled “Lysozyme and cancer: role of exogenous lysozyme as anticancer agent”, it can be stated, with some confidence, that HEWL modifies tumor growth because it activates immune responses, as shown in [27] and the references cited therein. A careful analysis of the studies on experimental models and even more of the studies, albeit few, on human tumors shows an indirect role in neoplasms. In other words, there is no direct inhibitory effect on the tumor cell [28], but instead an amelioration of neoplastic pathology, often documented by a specific effect on cells of the immune system, as paradigmatically highlighted by Vacca et al. in patients with multiple myeloma [29].

The role of HEWL as a chemotherapeutic agent against cancer is therefore questioned. We expect a chemotherapeutic agent to have some kind of direct effect against tumor cells, and we expect this effect to be documented both in vitro and in vivo. As we will see in the next subsection, the role of a direct effect of HEWL on tumor cells must be deferred in favor of a more complex mechanism of action in which host immunity plays a critical role (Figure 2). Of course, the proposal here is not to use HEWL as an anticancer agent, but to use tumor models to show how the interactions of HEWL with host immunity may be important in pathology.

### In Vivo HEWL Effects on Models of Tumor Growth

In the following, we focus on the modulations of neoplastic growth and spontaneous metastatic spread of two experimental mouse tumors, Lewis lung carcinoma and MCa mammary carcinoma of the CBA mouse, and briefly discuss the more immunogenic B16 melanoma.

Oral administration of HEWL at doses ranging from 25 to 200 mg/kg to animals with Lewis lung carcinoma results in a slight, often insignificant, reduction in tumor growth at the primary site and a pronounced and statistically significant reduction in the development of lung metastases. In this experimental model, HEWL is administered to the animals by mixing it with feed, taking into account the amount of feed consumed daily by each animal [30]. This study shows that the effect of HEWL is not due to a direct interaction with the tumor. This is because even when the tumor is implanted into animals that have taken HEWL orally prior to tumor implantation, the antimetastatic effect is equally clear and pronounced. Under these experimental conditions, it is unlikely that there is direct contact between the dietary-ingested HEWL and the metastatic tumor cells, since the metastatic process of this tumor begins about 5 to 7 days after implantation of the primary tumor, i.e., at a time when the HEWL that entered the bloodstream has presumably already been totally eliminated. Thus, the hypothesis that HEWL acts as a modulator of biological responses to metastatic growth is more than plausible (Figure 3A,B).

Confirmation of an indirect pharmacological effect of HEWL and, more precisely, of a biological response modifier effect, can be derived from two experiments [31]. In the first, animals carrying metastatic MCa mammary carcinoma were treated intravenously for 7 days with 100 μL of plasma from healthy animals that had been given 25–100 mg/kg/day of HEWL orally, as described above, mixed with the diet. In the second experiment, another group of animals carrying the same tumor was treated intravenously with mononuclear cells (mainly lymphocytes and monocytes) obtained from a peritoneal lavage of animals treated with HEWL as in the previous experiment (Figure 3C). In both cases, there was a significant reduction in metastasis formation in tumor-bearing animals treated with plasma or with cells derived from peritoneal lavage, although the effects on primary tumor growth were very small. When the primary objective of the study was the overall survival of tumor-bearing animals, the use of plasma samples collected from lysozyme-treated mice in conjunction with the surgical removal of the primary tumor increased the number of cured animals from 10% to 40% [32]. The more pronounced effect on metastases compared with the primary tumor suggests a possible involvement of the immune system. Indeed, it is more likely that immune defense is effective on single tumor cells during the process of spreading from the primary tumor and implantation in the target organ than in a primary tumor resulting from the forced transfer of a million cells in a single bolus.

This hypothesis is further supported by a study using B16 melanoma in mice. In an experimental setting in which mice were orally administered 50 mg/kg/day HEWL for 7 days after tumor implantation, a 66% reduction in lung metastases was observed independent of a rather nonsignificant reduction in tumor growth at its primary site. The effect on metastases corresponds to a significant increase in the lifetime expectancy of animals that had their primary tumor removed before treatment with HEWL with the same regimen (50 mg/kg/day for 7 days) [32].

## 3. HEWL Interactions with the Immune System

In a recent review of the potential therapeutic applications of lysozyme, the immunomodulatory properties of lysozyme, among others, were highlighted [16]. The authors’ analysis mainly referred to the properties of endogenous lysozyme in modulating some immune system responses, particularly involving lysozyme of macrophage origin. The purpose of the present review is rather to draw attention to the modulatory properties of immune responses of lysozyme from egg white (HEWL). In this context, the work of Namba and coworkers [33] is of much greater importance. They attempted to explain the mechanism of the immunomodulatory activity of HEWL administered orally in a guinea pig model by the production of peptidoglycans. According to this hypothesis, the bacteriolytic activity of HEWL is responsible for the degradation of the bacterial cell wall in the intestine, and the released peptidoglycans are ultimately responsible for the activation of immune responses after absorption into the bloodstream. This hypothesis is supported by the work of Bu and coworkers, in which the authors show that the bacterial fragments obtained by degradation with HEWL can elicit a pronounced immune response after administration to the experimental animal, suggesting the role of macrophage-dependent innate immunity through the activation of specific cathelicidins [34].

Analysis of data from experiments with tumors shows that HEWL significantly increases the number of immune cells both in the peritoneal cavity and in systemic circulation [31]. Moreover, the antitumor activity of HEWL cannot be simply limited to the soluble peptidoglycan fragments present in plasma, since immune cells harvested from the peritoneal cavity of HEWL-treated animals show antitumor effects comparable to those of HEWL-treated tumor-bearing animals. In other words, HEWL is capable of eliciting direct, probably nonspecific, stimulation of host immunity in addition to a peptidoglycan-mediated effect, which may lead to detectable pharmacological effects. Indeed, HEWL administered orally for 7 days at a dose of 25 mg/kg/day promotes an increase in peroxidase-positive cells in the lung, not restricted to neutrophils but probably due to newly recruited cells from the bloodstream, and a 1.3-fold increase in spleen weight compared with untreated animals [31]. The effect on the spleen seems particularly significant considering that the natural decrease in splenic lysozyme is an indicator of the decrease in immune responses associated with aging [35].

The work of Das and coworkers shows that lysozyme is capable of inducing lymphocyte responses against the development of tumors in experimental animals [36]. On the one hand, the authors postulate the possibility that immune stimulation results from modifications of lysozyme on the membrane of tumor cells, as previously suggested by Warren [37] and Osserman [38], causing recognition of altered cells by lymphocytes; on the other hand, similar stimulation of lymphocytes against tumor cells is also observed when treatment with HEWL occurred before the implantation of the tumors, clearly indicating a modulatory effect of the immune system independent of an effect on the surface of the tumor cells. The main effects of HEWL on immune effector cells and on immune structures in vivo and ex vivo described here and in the following sections are listed in Table 1.

### 3.1. Effects on Effector Cells

A 1993 paper reported that lysozyme is able to modulate lymphocyte responses depending on dosage, timing of administration, and lymphocyte cell status, suggesting that it can regulate lymphocyte proliferation in response to a variety of stimuli and the course of all cellular events involving the immune system that occur over time [49]. The results of the work of Valisena et al. [50] come to the same conclusions, pointing to the importance of HEWL dosage for the type of modulation of the immune response and to the specificity of the effect and the need to maintain the enzymatic activity of the molecule. An interesting review by Guryanova [51] highlights that metabolites and fragments of bacterial cells play an important role in the formation of immune homeostasis and that muramyl peptides possess a positive or negative regulation of inflammation depending on the microenvironment and duration of action. Recognition of peptidoglycans by cytosolic receptors NOD1 and NOD2 stimulates downstream pro-inflammatory signaling events via activation of NF- κB, including the production of pro-inflammatory cytokines such as interleukin-8 (IL-8) and antimicrobial molecules [52,53]. Further details can be found in a review by Ragland and Criss [54], which emphasizes the switch from bacterial cell killing to immune regulation. In any case, the antitumor activity of HEWL does not appear to be through the TNF pathway, and there is no evidence that it contributes to LPS (lipopolysaccharide) activity, while on the contrary it is able to stimulate mitogenic activity to ConA (concanavalin A) of ex vivo-isolated lymphocytes in the presence of IL-2 [41]. HEWL was found to be able to stimulate the recovery of the response to ConA of mononuclear cells reduced by tumor growth; this effect was particularly evident in splenocytes and GALT (gut-associated lymphoid tissue) lymphocytes. The effect of HEWL depended: (i) on the presence of plastic adherent cells; (ii) at least in the in vitro experiments, on the amount of HEWL used, which was higher at a dose of 250 μg/mL of incubation mixture; and (iii) on the timing of HEWL treatment.

However, it is also important to emphasize what we know about the regulation of its activity mediated by dendritic cells. HEWL can be processed by dendritic cells favoring the expression of type B peptide-MHC conformers depending on whether inflammatory processes are present or not [55]. In the absence of inflammatory processes, the phenomenon is almost irrelevant, whereas in the presence of inflammation, with the presence of type I interferon and toll-like receptor ligands, processing by macrophages increases significantly, leading to an increase in type B peptide-MHC conformers [56]. Therefore, it can be assumed that the effect of lysozyme is closely related to its dosage.

### 3.2. Effects on Immune Structures

Oral administration of 100 mg/kg/q1–9 days of HEWL is capable of restoring the tumor-growth-suppressed response of mononuclear cells to ConA in both spleen and GALT. In in vitro experiments with mononuclear cells from these two districts, the effect of HEWL was closely related to the dose used and the duration of contact with the cells, and was demonstrated both by measuring the incorporation of ^3^H-thymidine into DNA and by assessing protein synthesis with the sulphorodamine B assay. Accordingly, HEWL significantly altered the architecture of the intestinal mucosa in vivo by reducing lymph nodes in close proximity to the epithelial cells of the intestinal villi [47]. A simple conclusion is that mucosal immunity is the target for orally administered HEWL and that the effect of HEWL on immune cells and immunologic structures is closely connected with its dosage, similarly to what has been reported for immunological effects depending on its enzymatic activity. HEWL is able to correct the tumor-growth-impaired response of splenic lymphocytes derived from animals treated in vivo with 100 mg/kg for 9 consecutive days to ConA. In vitro, HEWL is capable of inducing proliferation activity in a mixed population of blast cells derived from mononuclear cells taken from the spleens of healthy animals treated as described above; HEWL is capable of eliciting a proliferative effect at IL-2, whereas similar cells derived from animals that did not receive HEWL were completely insensitive [48]. In the same paper, the statistically significant contribution to the cancer therapy by 5-FU (5-fluorouracil) is highlighted. Similar effects had already been observed with the model of Lewis lung carcinoma where HEWL significantly increased the curative effect of cisplatin [30]. Overall, these effects demonstrate the potential of orally administered HEWL on host immunity, suggesting that the immunomodulatory effect contributes significantly to the success of the therapy.

Under the same experimental conditions of 100 mg/kg/day in mice bearing MCa mammary carcinoma, HEWL increased the number of lymphocytes expressing the CD3, CD4, CD8, and CD25 antigens of lymphocytes in the intra-epithelial gut and mesenteric lymph nodes. In the mesenteric lymph nodes, the effect occurs after a few days of treatment, remains high throughout the treatment period, and returns to control levels after treatment is stopped. In this district, a decrease in the CD4:CD8 ratio in favor of CD8 lymphocytes is observed. In contrast, in the intraepithelial lymph nodes, the ratio between CD4+ and CD8+ cells, the percentage of CD3-positive lymphocytes, or lymphocytes expressing the IL-2 receptor do not change compared with controls. Under these experimental conditions, HEWL retains its enzymatic activity along the entire digestive axis, with approximately 10% of the administered dose still detectable in the jejunum one hour after treatment [42].

Overall, these results, together with those previously reported, support the ability of orally administered HEWL to elicit an immune response along the GALT-mesenteric lymph node axis against systemic inflammatory pathology caused by metastatic tumor growth.

The immunostimulatory effect of HEWL may have useful applications in the field of vaccines. In an experiment in which animals were given an antigen of Vibrio anguillarum and administered oral HEWL, a significant increase in lymphocytes in Peyer’s patches and, after a booster after three weeks, a significant increase in CD3-positive lymphocytes in mesenteric lymph nodes was demonstrated [57].

These data currently support the hypothesis that immunomodulatory activity is related to the presence of enzymatically active HEWL in the digestive tract. In other words, the modulation of immunity by HEWL is mediated by peptidoglycan fragments released by HEWL lysis of the bacterial cell wall in the intestinal lumen, the role of which has been extensively demonstrated in the past and also in the authors’ laboratory [33,58,59]. Studies with PEGylated HEWL administered orally, as in the previously described studies, show that immunomodulatory activity is maintained even when the product has completely lost its enzymatic activity on the peptidoglycan [43,44,60], supporting the thesis that HEWL as such is capable of producing pharmacologically active immunostimulation (Figure 4). However, it should always be considered that HEWL may cause other beneficial effects in addition to immunological effects. For example, the use of a HEWL coupled to polyethylen glycol has clearly demonstrated that the effect on tumor metastases, which has been abundantly demonstrated in various experimental models, has a direct effect on tumor cells in addition to stimulating immunity [45]. In vivo administration of doses acting on metastases causes a decrease in the adhesion molecules ICAM-1 and E-cadherin on tumor cells, which under these experimental conditions, in addition to controlling metastasis spread, are responsible for reducing lymphocyte infiltration of the primary tumor and decreasing the ability of splenocytes to bind to tumor cells [45]. It can be concluded that the effect of mPEG–lyso is due to a combination of effects on immunity and on tumor cells. In support to the effect of HEWL on the immune system, it is interesting to note that also the use of a recombinant human lysozyme coupled to the polyethylene glycol (rHLZ–PEG), in contrast to the same free form administered orally, is able to reverse the immunosuppression caused by cyclophosphamide and cyclosporine [60]. Also in this case, the rHLZ–PEG complex completely lost its antibacterial enzymatic activity, so the effect cannot be attributed to the formation of peptidoglycan fragments. Moreover, conjugation with polyethylen glycol is known to favor the intestinal absorption of the molecules to which it is bound, and this pharmacokinetic enhancement may explain the greater effect of rHLZ–PEG compared to free recombinant human lysozyme.

On the other hand, oral administration of HEWL is not limited exclusively to the intestinal lumen, as increasing plasma concentrations of this lysozyme can be detected throughout the treatment period within doses useful for stimulating innate immunity [61].

## 4. Study with HEWL in Patients

The effect of oral administration of 2 g/day of HEWL (HCl salt, medicinal product Lisozima SPA, Milano, Italy) was studied in a group of 23 patients who were in complete remission from malignant (21 patients) or inflammatory (2 patients) disease (Figure 5). Drug treatment began after a minimum of 60 days without supportive therapies (e.g., vitamins) and 4 to 6 months after the last administration of chemotherapy or radiotherapy. No other medication was administered during the study. The main effect of lysozyme was to restore the white blood cell count to the normal range of 4800–8500 cells/mm^3^, and the effect was particularly evident after 30 days of treatment. The main contribution to the normalization of the leukocyte count was due to neutrophils and lymphocytes [62]. Lysozyme treatment resulted in a reduction in CD8^+^ and Leu 11^+^ cells, with little or no effect on CD3^+^, CD4^+^, and Leu 7^+^ cells. The reduction in CD8^+^ cells favored the normalization of the CD4^+^:CD8^+^ cell ratio in 11 of the 16 patients evaluated. These effects can be attributed exclusively to lysozyme, since these values and, in general, the number of leukocytes returned to pretreatment values after lysozyme withdrawal. No correlation was found between these effects and the type of disease of the individual patient [46]. These data are consistent with the chemotaxis, leukocytosis, and increased lymphocyte reactivity observed in experimental models and described in the previous sections.

## 5. Conclusions

The discovery of lysozyme by Sir Alexander Fleming has just turned 100 years old. It is one of the model molecules in biochemical studies and is the most effective mucosal barrier for fighting bacterial infections. In recent years, its properties have been highlighted as effective, at least at the experimental level, in diseases such as COPD (chronic obstructive pulmonary disease) [63,64], diabetes [65], and viral infections [66,67,68,69,70,71]. However, one of the most studied and interesting properties is its ability to elicit immunological responses. The study of the interactions of lysozyme with the immune system mostly concerned its parenteral administration and the resulting reactions of a purely allergic nature, often using different types of lysozymes. In contrast, here we wanted to highlight the immunological properties of a specific lysozyme, hen egg white lysozyme (HEWL), to be administered orally, starting from the evidence published by Ragland and Criss [54] describing the transition of effects from killing bacterial cells to immunomodulation.

When administered orally, HEWL does not exhibit any toxicity, either local or systemic. The most interesting data at this level come from a small clinical trial in which all patients who received HEWL orally (usually 1 g in the morning and 1 g in the evening over a period of up to 180 days) reported no adverse events. Conversely, many patients asked about the possibility of continuing therapy beyond the study period, as they felt well and the colds typical of the winter season were shown to be reduced [46].

The study of the systemic effects of HEWL on the immune system in the experimental models studied shows the ability to activate responses along the GALT–systemic immunity axis. The effects observed on the lymphatic structures of the digestive mucosa subsequently move to the mesenteric lymph nodes and then to the systemic level, promoting pharmacological responses. Experimental data support the involvement of immune structures, such as the lymph nodes and spleen, as well as immune surveillance cell types. What is clear is that the immune stimulation cannot be attributed solely to the enzymatic property of the HEWL [72], as it is also evident when using an HEWL modified by PEG residues, which has lost the ability to cleave the bacterial peptidoglycan into immunoactive muramyl peptides. The mechanism of this activation is unclear.

Although the exact mechanism of immune response activation is not yet known, the beneficial systemic effects of this activation are clearly evident regardless of the study model used. Overall, therefore, the available data from the literature suggest that orally administered HEWL is capable of producing an effective immunostimulatory response in experimental animals and, albeit only in a small and specific cohort studied so far, in humans.

## Figures and Tables

**Figure 1 molecules-28-05027-f001:**
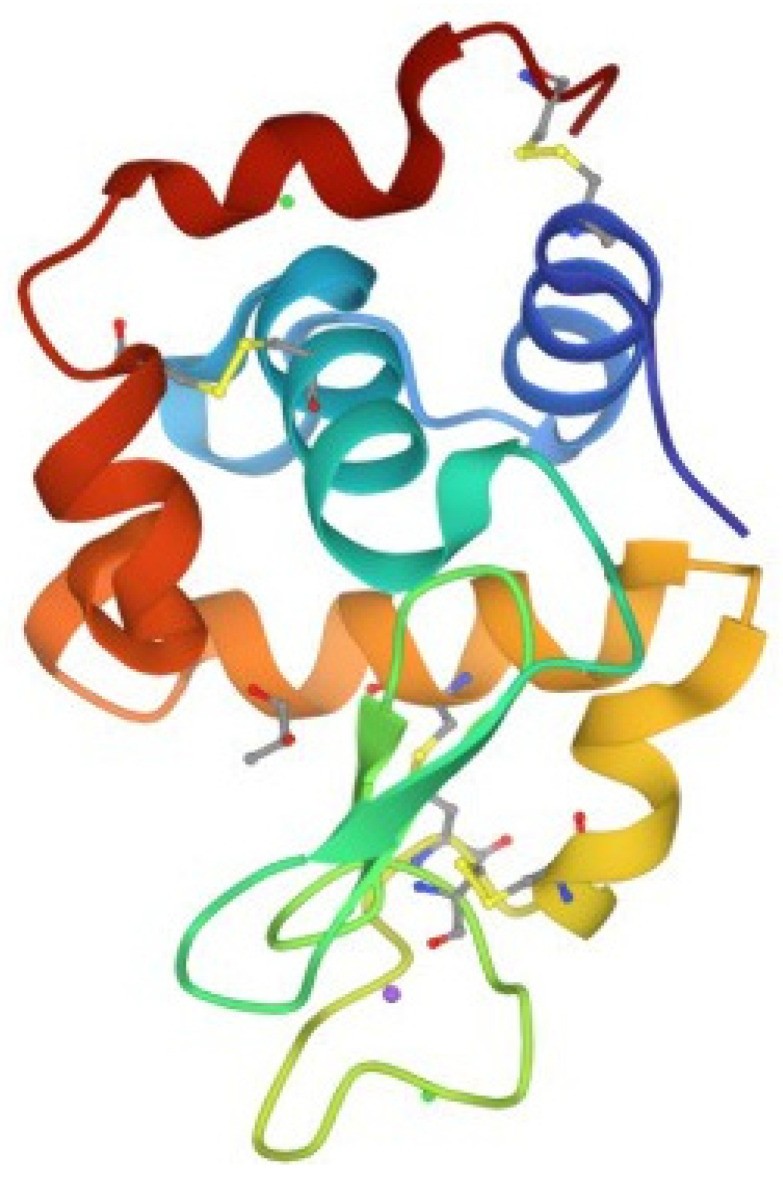
Structure of hen egg white lysozyme (HEWL). Lysozyme, also known as muramidase, is an antimicrobial protein that can hydrolyze the bacterial cell wall. Lysozymes are ubiquitous in the animal kingdom and have a similar overall structure. The figure refers to chicken or conventional type (c-type) lysozyme; it is a small single-chain protein with 129 amino acids [6].

**Figure 2 molecules-28-05027-f002:**
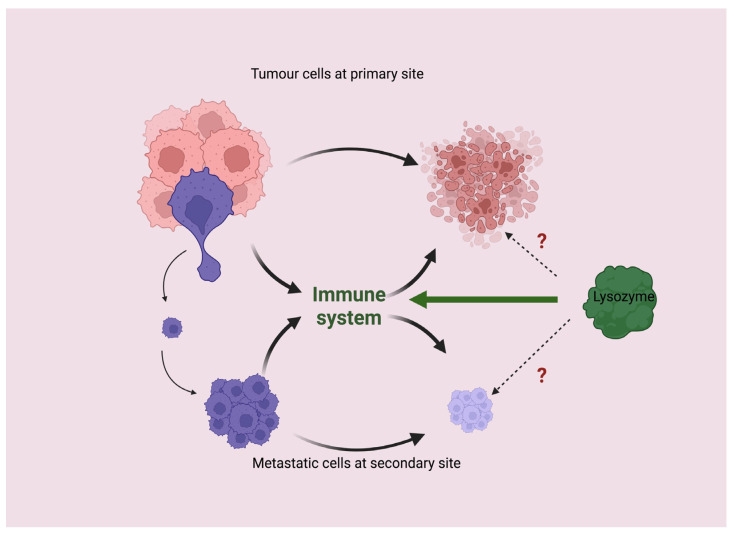
Hypothesis about the role of lysozyme as an agent in cancer treatment. A direct effect of lysozyme against tumor cells should be ruled out in favor of a more complex mechanism of action in which host immunity plays a crucial role.

**Figure 3 molecules-28-05027-f003:**
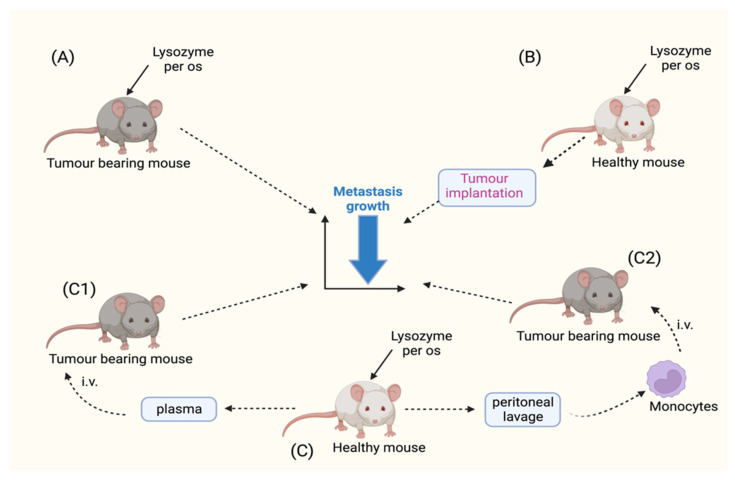
The role of exogenous lysozyme in experimental cancer models. Oral administration of HEWL (hen egg white lysozyme) to mice after (**A**) or before (**B**) tumor implantation with Lewis lung carcinoma [30]. HEWL was also administered orally to healthy mice (**C**) before plasma and peritoneal mononuclear cells were harvested for intravenous administration to two groups of mice with MCa mammary carcinoma ((**C1**) and (**C2**), respectively) [31]. In all experimental setups, the growth of primary tumors and metastases was examined, and similar effects were observed: a slight, often insignificant, reduction in tumor growth at the primary site and a pronounced and statistically significant reduction in the development of lung metastases, which are much more sensitive to immunomodulation because of their developmental and growth characteristics. These experiments highlight the critical role HEWL plays in triggering host immunity in in vivo cancer models. (See text for more details).

**Figure 4 molecules-28-05027-f004:**
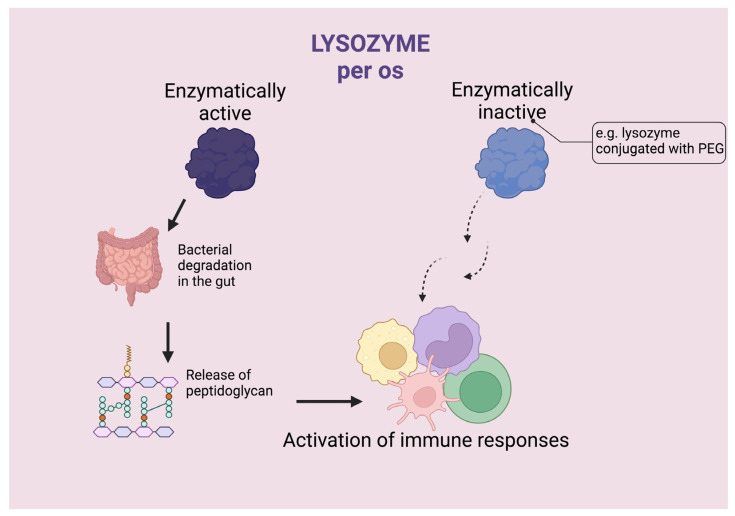
The role of the enzymatic activity of lysozyme in its interaction with the immune system. The immunomodulatory activity of lysozyme is related to the presence of enzymatically active lysozyme in the digestive tract (left) [33,34,58,59]. However, studies with pegylated lysozyme, both from hen’s egg white and recombinant human lysozyme, show that immunomodulatory activity is retained even after the product has completely lost its enzymatic activity on peptidoglycan [43,44,60].

**Figure 5 molecules-28-05027-f005:**
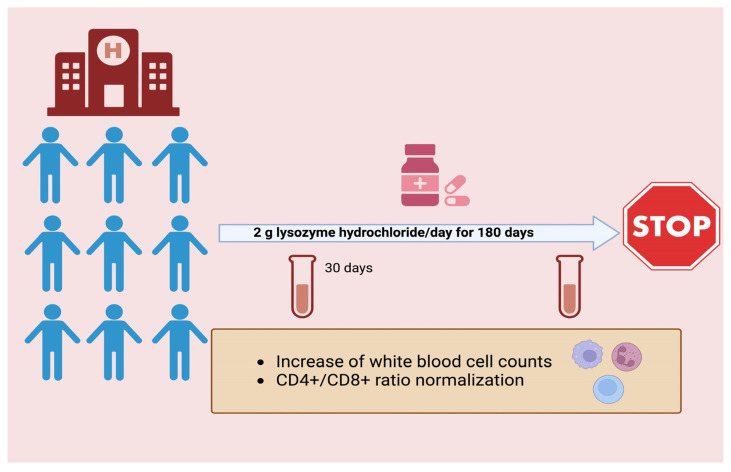
Study with lysozyme in patients. Lysozyme hydrochloride was administered orally for up to 6 months to patients who were in complete remission from malignant or inflammatory diseases [62]. The main effect of lysozyme was to restore the white blood cell count to the normal range.

**Table 1 molecules-28-05027-t001:** Effects of Lysozyme on immune effector cells in different experimental settings.

Experimental Conditions	Lysozyme Treatment	Murine Model	Immunomodulatory Responses	
In vivo	25 mg/kg/q1–7	Lewis lung carcinoma	Increase in peroxidase + cells in spleen (3.4× vs. healthy mice and 1.7× vs. tumor-bearing mice)	[39]
	25–100 mg/kg/q1–7	Healthy mice	43–47% increase in peritoneal resident cells 32–43% increase in circulating leukocytes	[31]
	100 mg/kg/q1–15	Lewis lung carcinoma	Increase in multinucleated giant cells of macrophage origin in the spleen	[40]
	100 mg/kg/q5–12	Lewis lung carcinoma MCa mammary carcinoma	Increased mitogenic response to ConA of ex vivo isolated splenocytes cultured in the presence of IL-2	[41]
	100 mg/kg/day/q1–9	MCa mammary carcinoma	Increase in lymphocytes from intra-epithelial gut and mesenteric lymph nodes expressing CD3, CD4, CD8, and CD25 antigens Reduction in CD4:CD8 ratio in favor of CD8+ cells in mesenteric lymph nodes	[42]
	350 mg/kg/q1–9 mPEG–Lyso ^	MCa mammary carcinoma	Increased response of lymphocytes to ConA and LPS vs. untreated controls	[43]
	350 mg/kg/q1–9 mPEG–Lyso ^	MCa mammary carcinoma	Recovery of decline in CD4+ lymphocytes caused by tumor growth Increase in number of lymphatic nodules in gut epithelium	[44]
	350 mg/kg/q1–9 mPEG–Lyso ^	TS/A adenocarcinoma	Decrease in infiltrating leukocytes in primary tumor	[45]
	2 g/day for 180 days	Patients affected by malignant or inflammatory diseases	Restoration of white blood cell count Reduction in CD8+ and Leu11+ cells Normalization of CD4+/CD8+ cell ratio in 11 out of 16 patients	[46]
Ex vivo	100 mg/kg/q5–12	Lewis lung carcinoma	Increased mitogenic response to ConA of ex vivo isolated splenocytes cultured in the presence of IL-2	[41]
	100 mg/kg/q1–10	MCa mammary carcinoma	Recovery of the tumor depressed response to ConA of splenocytes and lymphocytes harvested from GALT Increasing of 3H-thymidine incorporation into DNA and protein synthesis increase in mononuclear cells from spleen and GALT	[47]
	100 mg/kg/q1–8	MCa mammary carcinoma	Correction of the response to Con A of splenocytes isolated ex vivo	[48]
	350 mg/kg/q1–9 mPEG–Lyso	TS/A adenocarcinoma	Decrease in tumor cells expressing ICAM-1 and E-cadherin Reduction in tumor cells in synthesis and premeiotic phases	[45]
In vitro	25–250 μg/mL	Ex vivo-harvested splenocytes	Stimulation of proliferative activity of lymphocytes to IL-2 Induction of proliferation of blast cells	[48]

ConA: concanavalin A; GALT: gut-associated lymphoid tissue; LPS: lipopolysaccharide; q1–7: daily treatment from day 1 to 7 after tumor implantation or for 7 consecutive days in healthy mice; ^ 100 mg/kg HEWL equivalents.

## Data Availability

Data supporting the work and the discussion were from literature examination of published work.

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
