# Peer review of "Pharmacological Modulation of Host Immunity with Hen Egg White Lysozyme (HEWL)—A Review"

_molecules, 2023, doi:10.3390/molecules28135027_

Round 1

Reviewer 1 Report

Authors have suggested that orally administered HEWL is capable of producing an effective immunostimulatory response in experimental animals and, albeit only in a small and specific cohort studied so far, in humans. The title is simple. The abstract section is well-framed and have comprised all the significant findings of the manuscript. Other sections are well correlated and arranged properly. The writing style of the manuscript is simple to understand hence I believe that it will gain reader interest. Diagrams are well drawn with good pixel quality. The table is highly presentable with a clear footnote. The conclusion section is well-framed and has summarized all the highlights of the study. Hence, in my opinion the manuscript can be accepted in its current form.

The writing style of the manuscript is simple to understand hence I believe that it will gain reader interest.  Hence, in my opinion, the manuscript can be accepted in its current form.

Author Response

We thank the reviewer for the positive comments on the article.

Reviewer 2 Report

The authors have discussed about the role of hen egg-white-lysozyme (HEWL) in pharmacological modulation of host immunity. Additional points that can be considered are as following:

1. The novelty of the article should be clearly highlighted as number of excellent reviews have already been published on this topic.

2. More references from last few years should be added to improve visibility and quality of current work.

3. The search strategy used for the literature review should be indicated.

4. The manuscript should be carefully checked for typographical errors.

5. More figures and tables should be included to convey the key points of this review article.

Moderate editing of English language is required.

Author Response

  1. The novelty of the article should be clearly highlighted as number of excellent reviews have already been published on this topic.

We have included a sentence in the article to emphasise that, despite the great interest in this topic documented by numerous publications in the literature, lysozyme is mainly described in contexts and applications distinct from modulation of host immunity.

  1. More references from last few years should be added to improve visibility and quality of current work.

The revised manuscript now cites additional work published in recent years. The list of references was renumbered accordingly.

  1. The search strategy used for the literature review should be indicated.

The papers used for the preparation of the article and cited here were collected from the main scientific databases such as PubMed, Medline, and National Library of Medicine; we have added the information on the literature search in the article.

  1. The manuscript should be carefully checked for typographical errors.

We carefully checked the manuscript throughout and corrected the typographical errors. The English language was also edited when necessary.

  1. More figures and tables should be included to convey the key points of this review article.

The article includes 5 figures and a table that address various aspects relevant to describing the interactions of lysozyme with the immune system and cover the main points described in the text. This iconographic content appears exhaustive and balanced when the manuscript is considered in its entirety.

Reviewer 3 Report

Bergamo and Sava prepared a literature review on the effects of orally administered hen egg-white lysozyme on the immune system, with the special focus on the modulation of neoplastic growth and metastasis. The work is based largely on the authors' own papers (22 references out of 66 are self-citations), but this is justified by the authors' large contribution to the literature in question.

The work is interesting and worth publishing.

Author Response

(The authors gave the same response as above.)
